# Effect of KOH on the Energy Storage Performance of Molasses-Based Phosphorus and Nitrogen Co-Doped Carbon

**Iris Denmark [1], Samantha Macchi [1], Fumiya Watanabe [2], Tito Viswanathan [1] and Noureen Siraj [1,*,]**

[1] Department of Chemistry, University of Arkansas, Little Rock, 2801 S. University Ave, Little Rock, AR 72204, USA; isdenmark@ualr.edu (I.D.); spmacchi@ualr.edu (S.M.); fxwatanabe@ualr.edu (T.V.)

[2] Center for Integrative Nanotechnology Sciences, University of Arkansas, Little Rock, 2801 S. University Ave, Little Rock, AR 72204, USA; txviswanatha@ualr.edu

\* Correspondence: nxsiraj@ualr.edu

**Abstract:** In this study, we have evaluated the effect of potassium hydroxide (KOH) on the energy storage performance of metal-free carbon-based materials prepared from molasses. Molasses are a renewable-resource biomass and economical by-product of sugar refinement, used here as a carbon precursor. Two co-doped carbon materials using molasses were synthesized via a time and cost-efficient microwave carbonization process, with ammonium polyphosphate as a phosphorus and nitrogen doping agent. The phosphorus and nitrogen co-doped carbon (PNDC) samples were prepared in the presence and absence of a chemical activating agent (KOH), to study the role of chemical activation on PNDCs. Physical characterizations were performed to gain insight into the composition, pore size and topographical data of each material. Electrochemical characterization via cyclic voltammetry in 1 M sulfuric acid ($H_2SO_4$) as well as in 6 M KOH as electrolytes, revealed high current density and specific capacitance for the chemically activated material (PNDC2) compared to one without chemical activation (PNDC1). The capacitance value of 244 F/g in KOH electrolyte was obtained with PNDC2. It is concluded that addition of KOH prior to carbonization increases the surface functionality, which significantly enhances the electrochemical properties of the PNDC material such as current density, stability, and specific capacitance.

**Keywords:** co-doped carbon; electrical double layer capacitor; cyclic voltammetry; pseudocapacitance; supercapacitor

## 1. Introduction

Currently, fossil fuel consumption drives our global economy. The non-renewable resource materials are the primary source of all energy consumed around the world for various purposes, from transportation to heating our homes. To emphasize the gradual declination of these precious resources, a 2009 study by Shafiee et al. [1] demonstrated the fossil fuel availability trends with respect to time. A prediction formula was established during this study that was based on the Klass model [2], which relates fossil fuel consumption reserves to the prices of oil, coal, and gas for time intervals of 35, 37 and 107 years. The equation demonstrated that coal reserves will most likely last until the year 2112 and will be the only fossil fuel remaining after 2042. However, gas hydrate reservoirs are the potential source of clean energy which has been currently explored that will remain after this point [3,4]. This leaves a pertinent need for energy storage devices, from alternate energy sources such as wind and solar. Batteries, and supercapacitors represent two important mechanisms for energy storage.

Supercapacitors are energy storage devices that have quickly grown in interest as a potential solution to the fossil fuel crisis. They have many applications including their use in transportation, small electronics (computers, tablets, etc.), and medical equipment, as they allow very rapid charging and discharging with high cycle lifetime. These devices

primarily operate on two phenomena, namely electric double layer capacitance (EDLC) and pseudocapacitance (PC). EDLC involves the electrostatic storage of charge (such as in a capacitor) while PC describes the electrochemical storage of energy governed by reversible redox reactions between electrode and electrolyte (such as in a battery) [3]. Both mechanisms contribute to the capacitance of supercapacitor devices and are greatly influenced by the nature of the electrode material chosen.

Many different kinds of materials have been explored as supercapacitor electrode materials such as activated carbons (ACs) [5,6], conductive polymers [7,8], and metal oxides [9,10]. Though conducting polymers and metal oxides exhibit exceptionally high capacitance values, they suffer from low cycling life, potential toxicity, and expensive synthesis [11]. Thus, carbon is one of the most promising choices as an electrode material because it is abundant, inexpensive, and can be produced in any form, from bulk to nano. Some of these materials include graphene [12], carbon nanotubes (CNTs) [13], quantum dots [14], and ACs [15,16]. ACs are highly promising, as they utilize a facile synthesis protocol to yield high-surface-area carbon products. These are often synthesized by chemical activation with an activating agent such as potassium hydroxide (KOH) [17], zinc chloride ($ZnCl_2$) [18], or sulfuric acid ($H_2SO_4$) [19]. Traditionally this is a two-step process in which biomass is first carbonized and subsequently activated. Recently, Serafin et al. [20] used one step method to chemically activate the carbon material with co-doping agents. However, extensive pre-preparation of carbon precursor is needed for this method.

Doping (of carbon) refers to the covalent incorporation of heteroatoms in the graphite/graphene structure. Doping has been found to enhance or tune the functionality of the material. Doped carbon is very useful as a supercapacitor electrode material and acts as a sponge for energy storage due to its porous and functional surface which can vary in size [21,22]. Recent works have also demonstrated that doping carbon materials with heteroatoms can further enhance their capacitive behavior [23,24]. Co-doped materials can possess high energy densities and can be made in a plethora of ways ranging from pyrolysis [25] to time-efficient microwave carbonization [26,27]. In addition to doping, surface area [28], pore size [29] and surface functionalities [30] are the key parameters for supercapacitor performance. A graphical construct created by Macchi et al. [31] compared the physical characteristics of four co-doped carbon materials (i.e., elemental composition and surface area) side by side with specific capacitance. All materials were compatible in capacitance, but demonstrated much variety in other key characteristics.

Herein, we report a phosphorus and nitrogen co-doped carbon (PNDC) derived from the microwave-assisted carbonization of molasses in presence of ammonium polyphosphate (APP). The product obtained by microwave-assisted carbonization of this simple mixture, will be referred as PNDC1. The addition of small amounts of KOH to the precursor mix prior to carbonization resulted in a chemically activated PNDC2 material. Potassium naturally occurs in molasses, and further addition in the form of KOH provides enhanced electrochemical characteristic of the material when carbonized (PNDC2). Synthesis of this specific material encompasses a microwave carbonization method that is highly time-efficient as compared to other methods employed for synthesis of carbon-based supercapacitor materials. In addition, KOH chemical activation occurred simultaneously with co-doping in a single step microwave environment by direct incorporation of activating agent with precursor materials which is the novelty of this work. The detailed physical characterizations are performed to explore different parameters such as pore size, surface area, elemental composition, and surface functionalities, which are very crucial for the performance of supercapacitor materials. The electrochemical characterizations are studied in acidic and basic electrolyte to investigate their specific capacitance values.

## 2. Experimental

### 2.1. Materials

Molasses (Grandma's (registered trademark) was purchased from a local grocery store. APP (MW = 97 g mol⁻¹) was a generous gift from CheMarCo Inc. (Greenville, SC, USA) KOH, $H_2SO_4$, and 99% ethanol were purchased from Sigma-Aldrich (St. Louis, MO, USA). Carbon black was obtained from Cabot Chemical (Boston, MA, USA). 5% Nafion D-520 was purchased from Bean Town Chemical (Hudson, NH, USA).

### 2.2. Synthesis of P and N Dual-Doped Carbon from Molasses

PNDC synthetic procedure involved the microwave assisted carbonization of molasses, a bioavailable, viscous syrup- and APP which served as a microwave absorber and the primary source of phosphorus and nitrogen dopant. PNDC1, as shown in Scheme 1, was synthesized by placing a combination of molasses and APP as, given in Table S1 in the supporting information, in a boron nitride (BN) crucible, with a BN cover on top and placed into a microwaveable furnace which was operated in a conventional microwave oven (Panasonic (Kadoma, Osaka, Japan)) for 30 min at 2.45 GHz and 1250 Watts of power. After cooling to room temperature, the resulting PNDC1 was obtained as a chunky, black solid, which was powdered with a mortar and pestle, to homogenize the carbonized sample. PNDC2 was synthesized similarly, but with the addition of 0.1 mL of 3 M KOH to molasses and APP mixture prior to carbonization. As mentioned in Table S1 in the supporting information, an increase in product yield for PNDC2 (compared to PNDC1) by 81% (from 0.165 g to 0.204 g) (Table S1) was observed. This is most likely due to the formation of oxidized carbon (oxygen doping) during carbonization in presence of KOH.

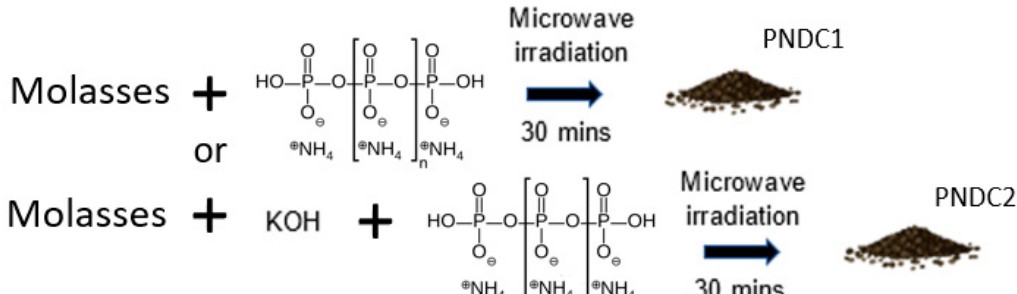

**Scheme 1.** Synthesis scheme of phosphorus and nitrogen co-doped carbon (PNDC1) and phosphorus and nitrogen co-doped carbon simultaneously co-doped and activated with potassium hydroxide (KOH) (PNDC2) from molasses. PNDC—phosphorus and nitrogen co-doped carbon.

### 2.3. Physical Characterization Methods

Both PNDC materials were characterized using different techniques to investigate the surface morphology, porosity and elemental composition, which plays a significant role in the performance of charge storage characteristics of the material.

PNDC's surface area and porosity were determined by correlating surface area to the amount of inert nitrogen gas adsorbed to the sample's surface. For this application, a Micrometrics (ASAP-2020) (Ottawa, ON K1P 5J2, Canada) surface area analyzer was used. Sorption studies utilizing Brunauer–Emmet–Teller (BET) analysis was conducted with nitrogen gas at a 77 K bath temperature in which approximately 0.15 g of sample weighed into a round bottom sample tube was used. Prior to analysis, both samples were washed via gravity filtration with 0.1 M hydrochloric acid (HCl) solution until neutral and oven-dried at 70 °C for 4 h. Samples were degassed for approximately 10 h prior to analysis.

Scanning electron microscopy (SEM) was performed using JEOL 7000F (Peabody, MA, USA) instrument to observe the surface morphology of samples. PNDCs were loaded onto an aluminum sample mount with double sided carbon tape prior to imaging. The same instrument was used for energy dispersive X-ray spectroscopy (EDS), a method that provides bulk elemental composition of the co-doped samples.

A Thermo Scientific (Waltham, MA, USA) K-alpha X-ray photoelectron spectrometer (XPS) was used to analyze the surface elemental composition and potential functional groups on the surface of PNDC materials. Survey and narrow scan analysis were conducted and analyzed utilizing Thermo Scientific Advantage software.

Raman spectroscopy was used to confirm additional morphological characterization regarding the nature of carbon in the materials. The powder sample was pressed onto a glass slide and analyzed at room temperature using a Horiba Jobin Yvon (Edison, NJ, USA) LabRam HR800 with a charge-coupled detector and spectrometer with a grating of 600 Lines mm$^{-1}$, excitation of 632 nm from a He–Ne laser with 10 mW intensity.

Amorphous nature and further surface characterization of PNDC2 was established using X-ray diffraction (XRD) studies. The sample was analyzed using a Bruker (Billerica, MA, USA) D8 Discovery XRD spectrometer. Results are shown in Figure S1 [32,33].

*2.4. Electrochemical Methods*

Cyclic voltammetry was conducted using an BASi Epsilon (Mt. Vernon, IL, USA) potentiometer with a traditional three-electrode setup. Two different aqueous electrolytes, acidic (1 M $H_2SO_4$) and basic (6 M KOH), were used in this study. The working electrode for cyclic voltammetry was a glassy carbon electrode coated with active material (PNDC) and a platinum wire served as a counter electrode. Reference electrodes utilized were Ag/AgCl and Hg/HgO for acidic and basic electrolyte, respectively. Nitrogen gas was purged through electrolyte solutions for 20 min prior to measurements to ensure inert conditions. To load the working electrode with active material, a slurry suspension was prepared in ethanol containing PNDC, carbon black, and Nafion in a 90:5:5 weight ratio. A 20 μL sample of this suspension was loaded onto glassy carbon electrode and dried prior to measurements. All electrochemical preparation and experimentation were performed at room temperature. Cyclic voltammograms were recorded from 0–800 mV and −1000–0 mV in acidic and basic media, respectively. Cyclic voltammograms were recorded at various scan rates from 5–100 mV/s. Electrochemical stability measurements were performed for 3000 cycles using cyclic voltammetry at 50 mV/s scan rate.

**3. Results**

*3.1. Physical Characterization*

3.1.1. Brunauer–Emmet–Teller (BET) Analysis

Comparative nitrogen adsorption and desorption isotherms were acquired for samples which were prepared as described in experimental section (Figure S2) indicate that both materials exhibit a Type IV isotherm behavior. Type IV isotherms are characterized by the occurrence of capillary condensation, indicated by a hysteresis loop through higher relative pressure range. Figure 1 demonstrates the pore size distribution of PNDC1 and PNDC2. PNDC1 shows to have a wider pore size distribution, whereas PNDC2 shows a pore size range that is larger in diameter and low in volume. These deeper pores in PNDC1 contribute to its greater surface area.

From these measurements several parameters are elucidated, such as BET surface area (SA), pore volumes, and pore distribution (micro- and mesoporosity). As per Table 1 for each material, $V_{micro}$ refers to the micropore volume, $V_{meso}$ refers to mesopore volume and $V_t$ refers to total pore volume. PNDC2 shows to be much more mesoporous, with a mesopore percentage of over 80% of total porosity as compared to 61% mesoporosity for PNDC1. Thus, it is concluded that chemical activation of the PNDC sample affects the pore volume and surface area of the sample as shown in Figure 1.



Usually, high surface area (SA) samples are considered to be ideal for supercapacitor application. However, in this study chemical activation of the PNDC sample before carbonization significantly reduces the surface area of the sample. Prolonged activation time has been known to decrease surface area due to increased diffusivity of the activating agent (KOH in this case) throughout PNDC matrix which compromises pore structure and consequently affects surface area [34–36]. This was compensated by increased surface functionality.

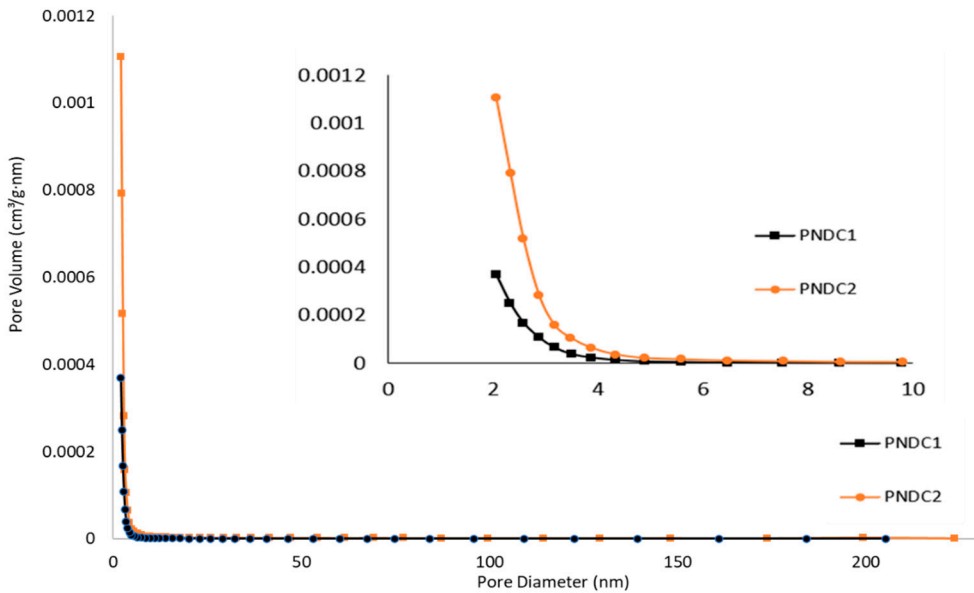

**Figure 1.** Pore size distribution plots for PNDC1 and PNDC2.

**Table 1.** Surface area and pore volume of PNDC1 and PNDC2 as obtained by Brunauer–Emmet–Teller (BET) analysis.

| Material | SA (m²/g) | $V_{micro}$ (cm³/g) | $V_t$ (cm³/g) | $V_{micro}/V_t$ (%) | $V_{meso}$ (%) | Average Pore Size (nm) |
|----------|-----------|---------------------|---------------|---------------------|----------------|------------------------|
| PNDC1 | 417 | 0.0645 | 0.164 | 39 | 61 | 2.194 |
| PNDC2 | 110 | 0.00663 | 0.0403 | 16 | 84 | 2.173 |

### 3.1.2. Scanning Electron Microscopy (SEM)

SEM imaging allows for a detailed analysis of surface morphology of PNDC materials. The morphology of both materials is unique to the carbonization process (Figure 2). At lower magnifications, PNDC1 demonstrates more of a 2-dimensional, rugged and speckled surface with crater-like features around 10 μm in diameter scattered across its surface. While at higher magnifications disc-like decorations planted into the surface were observed that are around 2 μm in diameter. However, chemically activated material PNDC2 presents more of a raised, complex, and rigid surface at higher magnifications and presents much larger crater-like features on the surface as big as 50 μm in diameter.

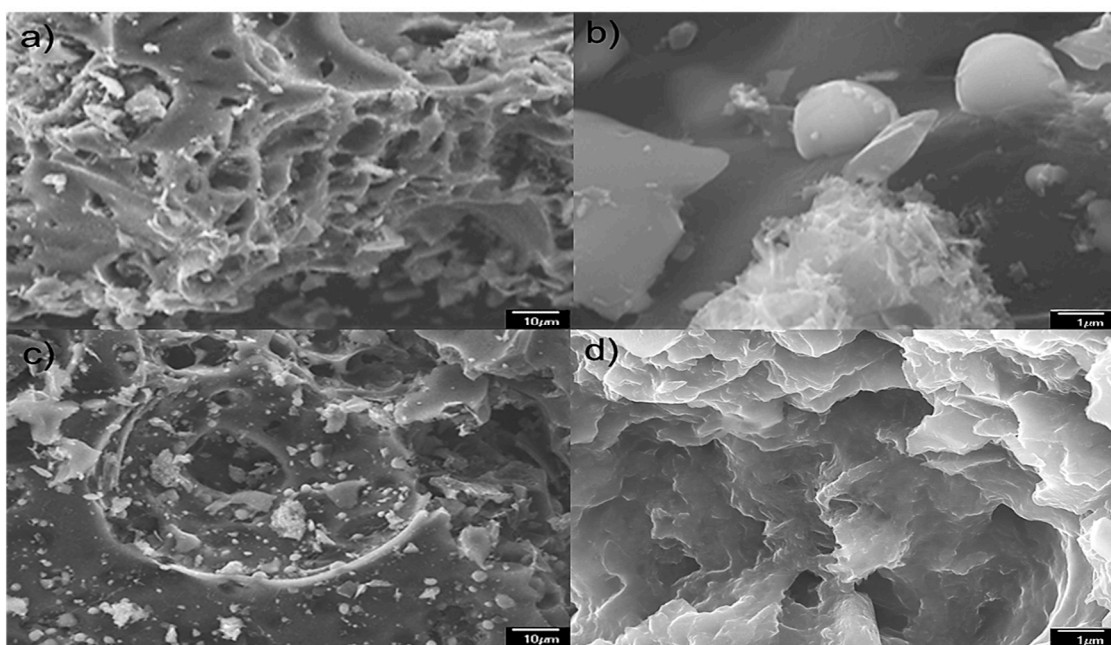

**Figure 2.** SEM images of PNDC1 (**a**,**b**) and PNDC2 (**c**,**d**). The left two images are of magnifications of the carbonized materials at 1000× magnification. The right two images illustrate 10,000× magnification.

### 3.1.3. Energy Dispersive X-ray Spectroscopy (EDS)

By utilizing EDS, the bulk elemental composition of PNDC1 and PNDC2 were determined. From the results demonstrated in Table 2, both materials were rich in carbon with notable concentrations of nitrogen and phosphorus. However, PNDC2 contains surprisingly low bulk percentage of nitrogen (~0.1 At %). Nitrogen groups that are initially present such as amines are unstable at high temperatures and if not transformed into more stable groups such as pyrrole, may decompose into nitrogen gases when KOH is present, while KOH activation concurrently induces carbon oxidation, which supports the high oxygen content of PNDC2 [37–39]. The essential differences in composition between PNDC1 and PNDC2 are mainly due to each material undergoing different reaction pathways during carbonization, especially considering the involvement of KOH in reaction and co-doping with surface oxygen during microwave irradiation. In this manner, KOH reacts with the environment under high temperatures to form potassium oxide ($K_2O$). $K_2O$ then reacts further with carbon and the low-pressure environment to form potassium carbonate ($K_2CO_3$) and carbon gases [40,41]; it is likely that nitrogen is reacted with excess oxygen to form nitrogen oxide gas during decomposition.

**Table 2.** EDS shows bulk elemental composition of both phosphorus and nitrogen co-doped materials (At%).

| Element | PNDC1 | PNDC2 |
|---------|-------|-------|
| C-K | 59.99 | 72.5 |
| N-K | 34.48 | >0.1 |
| P-K | 0.75 | 4.98 |
| O-K | 4.47 | 20.1 |
| K-K | >0.1 | 0.32 |

### 3.1.4. X-ray Photoelectron Spectroscopy (XPS)

It is important to consider the surface elemental composition of a supercapacitor material because most electrochemical reactions occur between electrolyte and functional groups at the material's surface. XPS allows us to analyze the elemental composition and potential functional groups present on the surface of the co-coped materials. From survey

scan data, both PNDCs contain similar atoms on the surface as in the bulk: carbon, oxygen, nitrogen and phosphorus. PNDC2 contains much greater potassium concentration on the surface (4.23%) than PNDC1 (Table 3), (due to addition of KOH during synthesis of PNDC2 that was not used in the synthesis of PNDC1). Additionally, PNDC2 contains a very small percentage of carbon (12.32%) with a relatively high amount of oxygen (64.37%) due to surface oxidation by KOH during the activation process [42,43].

Narrow scan studies were performed to determine elemental bonding information for PNDC samples. Table 4 depicts the composition of various functional groups present on the surface of two materials. PNDC1 and PNDC2 exemplify mainly sp2 hybridized carbon bonding with a correlating binding energy of 285 eV (Figure S3). PNDC1 also showed C1s peak located at 286 eV which is attributed to C-O binding on the material's surface. A doublet is observed at values 293 eV and 295 eV for PNDC2 that is very likely due to the presence of potassium. The photoelectric effect from XPS processing causes splitting of potassium into its 2p3/2 and 2p1/2 states respectively [44]. There are four different nitrogen bonding environments found in the XPS at 398, 400, 401, 403.5 eV correlating to pyridinic, pyrrolic, graphitic, and nitrogen oxide bonding [45,46]. Both PNDC1 and PNDC2 exhibited pyridinic nitrogen at relatively small amounts (0.5 and 0.44%, respectively). PNDC1 contains a significant amount of pyrrolic nitrogen at 1.43% while PNDC2 exhibits primarily the presence of organic (graphitic) nitrogen (1.57%). Oxygen containing groups were found at 531, 533, and 535 eV which are correlated to carbonyl, quinone and ether groups respectively [47,48]. Most oxygen functional groups for both PNDC1 and especially PNDC2 were present in the form of quinone and oxygenated carbonyl groups. Two phosphorus environments were located at 134 and 135 eV which are attributed to organic phosphate (POC) and phosphorus oxide functional groups respectively [33,34]. Both PNDC1 and PNDC2 exhibit POC contribution but only PNDC2 has P-oxide groups at the surface in a relatively high amount (13.16%). Overall, PNDC2 has a higher percentage of functional groups present on the surface, which can prove to be significant towards pseudocapacitance contribution during electrochemical performance.

**Table 3.** Survey scan results of C1s, N1s, P2p, K2p and O 1s core level XPS spectra of PNDCs.

| Sample | C1s (At%) | N1s (At%) | P2p (At%) | K2p (At%) | O1s (At%) |
|--------|-----------|-----------|-----------|-----------|-----------|
| PNDC1  | 66.24     | 5.10      | 4.93      | >0.01     | 22.95     |
| PNDC2  | 12.32     | 0.96      | 17.79     | 4.23      | 64.37     |

**Table 4.** Detailed narrow scan XPS analysis of functionalities on PNDC surfaces. (At %).

| Element: Functionality | PNDC1 | PNDC2 |
|------------------------|-------|-------|
| O1: quinone | 4.42 (530.42 eV) | 12.43 (531.36 eV) |
| O2: carbonyl | 8.16 (532.38 eV) | 40.44 (532.55 eV) |
| O3: ether | 1.74 (534.97 eV) | ~ |
| P1: organic phosphorus | 0.44 (135.80 eV) | 2.91 (135.10 eV) |
| P2: P-oxide | ~ | 13.16 (134.38 eV) |
| N1: graphitic | ~ | 1.57 (401.12 eV) |
| N2: pyridinic | 0.5 (397.92 eV) | 0.44 (398.31 eV) |
| N3: N- oxide | 0.34 (403.51 eV) | ~ |
| N4: Pyrrolic | 1.43 (400.24 eV) | ~ |

### 3.1.5. Raman Spectroscopy

Raman spectroscopy allows for insight into the characterization of PNDC carbon bonding environment. The extent of material disorder is illustrated by presence of a disorder (D) peak in the Raman spectra, whereas ordered graphitic sp2 carbon binding is characterized by presence of a graphitic (G) band. The spectrum for PNDC1 and PNDC2 displays two distinct Raman peaks (Figure 3) at Raman shift values of around 1350 cm$^{-1}$ and 1600 cm$^{-1}$ respectively for D and G peaks which are characteristic of heterogeneous carbonaceous materials [49,50]. This mode is forbidden in perfectly ordered graphite. The G peak, also known as the graphitic band, is representative of sp2 carbon pair stretching modes. The ratio of these two peaks, $I_D/I_G$, provides information related to the degree of doping for the sample. PNDC2 has a defect ratio of 2.6 as compared to PNDC1's defect ratio value of 1.8. This is indicative of a higher doping effect in PNDC2. This result is well supported by previously observed elemental composition data acquired from XPS and EDS.

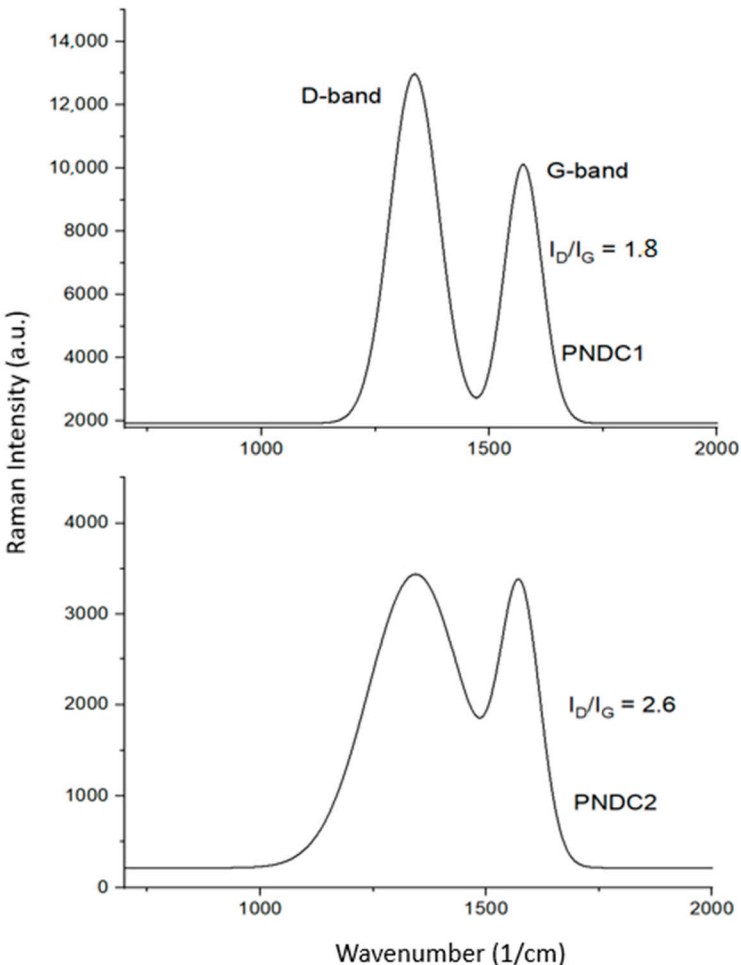

**Figure 3.** Raman spectra for PNDC materials.

### 3.2. Electrochemical Analysis

The resulting molasses-derived, PNDC materials have been investigated for electrochemical performance using cyclic voltammetry in both acidic and basic media. Specific capacitance was determined using the following Equations (1) and (2):

$$I_{avg} = |(I_a + I_c)| \div 2 \qquad (1)$$

$$C_s = I_{avg} \div (m \times \nu) \tag{2}$$

where average current is symbolized by $I_{avg}$ and peak currents are given by symbol 'I' in which $I_a$ is the anodic and $I_c$ correlates to cathodic peak current. 'm' is the mass of active material on the electrode and $\nu$ is scan rate (V/s).

In acidic electrolyte (1 M $H_2SO_4$), voltammogram shapes for PNDC1 and PNDC2 are generally rectangular which is a characteristic of EDLC charge storage. PNDC1 exhibits slight vertical widening at around 0.35 V (Figure S4) which is indicative of pseudocapacitance-specific (electrochemical) behavior [49]. Both materials exhibited sufficient surface elemental doping that contributed to the presence of functional groups such as pyridinic and quinone groups that allow electrochemical redox capability of these materials. Maximum specific capacitance values for PNDC1 and PNDC2 in 1 M $H_2SO_4$ were calculated to be 143 and 219 F/g respectively at 5 mV/s scan rate. The greater value for PNDC2 is attributed to the enhanced surface functional group composition compared to PNDC1. There were uniform trends in reactivity for the varying scan rates of each material. In basic electrolyte (6 M KOH) voltammograms for both samples are quasi-rectangular in shape (Figure 4), similar to results in acidic environment. However, there are no notable redox peaks observed for either sample in 6 M KOH electrolyte unlike in acidic electrolyte (Figure S5), indicating that it is purely EDLC mechanism in charge storage as compared to the reported voltammograms of PNDC1 and PNDC2 in acidic conditions, as discussed above [51]. At all scan rates PNDC2 exhibited a greater current density than PNDC1. Thus, PNDC2 also demonstrated significant electrochemical enhancement with a specific capacitance of 244 F/g compared to that of 157 F/g reported for PNDC1 in 6 M KOH electrolyte. Thus, it is concluded that PNDC2 showed the better charge storage properties in both acidic and basic electrolyte due to chemical activation.

The highest specific capacitance values for both samples was reported at 5 mV/s scan rate in both acidic and basic media. The capacitance performance decreases as the scan rate is increased, due to the increased ohmic resistance of ions during material diffusion and less contact with functional groups occupying PNDC surface and pores, essentially lessening the effect of energy storage [52,53]. Figure 5 highlights this inversely proportional relationship between scan rate and resulting specific capacitance for PNDC1 and PNDC2 in acidic and basic electrolyte. In acidic media both PNDC1 and PNDC2 experience losses in specific capacitance from 5 to 100 mV/s (17% and 32%, respectively). In KOH electrolyte, PNDC2 comparably suffers a greater loss (42%) compared to PNDC1 (26%) with increase in scan rates. This indicates that the electrolyte species can better interact with PNDC1 surface at higher scan rates.

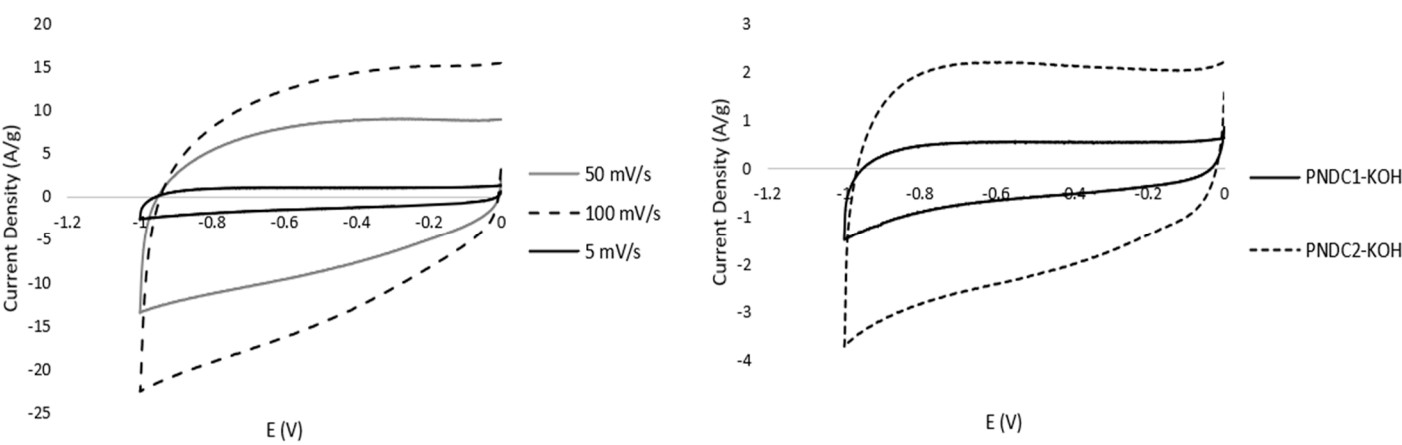

**Figure 4.** Cyclic voltammograms in 6 M (KOH) for PNDC2 at varying scan rates (**left**) and PNDC1 compared to PNDC2 in 6 M KOH electrolyte at a scan rate of 5 mV/s (**right**).

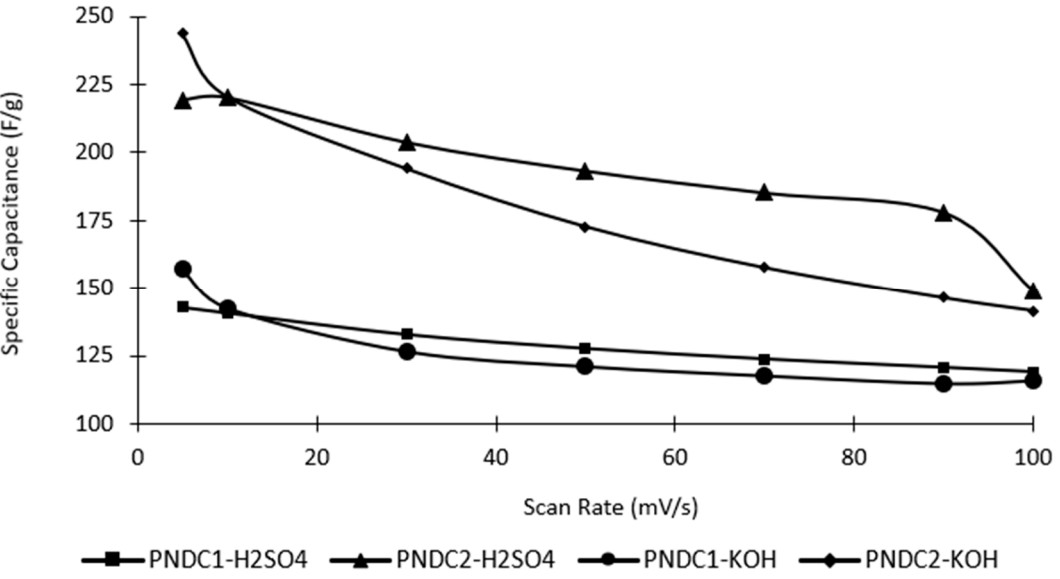

**Figure 5.** Plot demonstrating dependence of specific capacitance on scan rate vs. 1 M sulfuric acid ($H_2SO_4$) and 6 M KOH.

Recyclability

Recyclability was performed to ensure the long cycle life of these kinds of PNDC materials. The molasses-based PNDC presented herein demonstrated good stability. As demonstrated in Figure S6, PNDC2 shows a 16% increase in current density with respect to each cycle for up to 3000 cycles. This may be due to a delayed activation effect in which the current density of the materials is gradually increased until the material is fully activated as a result of electrochemical charge-discharge reactions and increase in wettability at the electrode/electrolyte interface [53–55]. This is largely due to reactions from the previously discussed prominent amount of oxidizing functional groups present on this material's surface. The gain in specific capacitance value from 1 cycle to 3000 cycles indicates that these materials are highly stable in basic electrolyte and thus are highly promising materials for application as supercapacitor materials.

## 4. Conclusions

The PNDC material synthesized by facile microwave-assisted method demonstrates ideal physical and electrochemical properties. The carbon precursor utilized in its synthesis is molasses, which is inexpensive because it is a renewable carbon-rich byproduct of sugar processing. We demonstrated that chemical activation of the samples can be performed prior to carbonization using a very facile and cost-effective microwave method. These newly developed materials exhibit high surface areas, which is a very important parameter for charge storage characteristic. Addition of KOH prior to the co-doping carbonization process tends to have a very direct effect on the PNDC's yield, composition, porosity, surface area and specific capacitance. Stability of PNDC2 is positively affected by high oxygen content. The higher percentage of functional groups at the surface of PNDC2 provided additional sites for superior supercapacitor performance. Addition of the KOH treated sample prior to carbonization proved to be a simple but elegant method to obtain high performance supercapacitor materials prepared by the microwave-assisted method.

**Supplementary Materials:** The following are available online at www.mdpi.com//2/1/3/s1, Table S1: Composition of PNDC1 and PNDC2 prepared prior to carbonization, Figure S1: X-ray diffraction spectra for PNDC-2., Figure S2: BET isotherm plot for PNDC1, Figure S3: XPS narrow scan single trace views of PNDC2 (left) vs. PNDC2 (right). XPS spectra are of C1s, O1s, P2p and N1s., Figure



S4: Cyclic voltammogram comparison in 1M H₂SO₄ electrolyte for PNDC1 (left) and PNDC2 (right) at varying scan rates., Figure S5: Cyclic voltammograms in 6M KOH electrolyte for PNDC1 (left) and PNDC2 (right) at varying scan rates., Figure S6: PNDC2 stability for 3000 cycles at 50 mV/s vs. 6M KOH electrolyte using three-electrode system.

**Author Contributions:** Conceptualization, I.D. and N.S.; methodology, I.D., N.S., S.M. and T.V.; software, I.D. and S.M.; validation, I.D.; formal analysis, I.D.; investigation, I.D.; resources, N.S. and F.W.; data curation, I.D. and F.W.; writing—original draft preparation, I.D., S.M. and N.S.; writing—review and editing, I.D., N.S., S.M. and T.V.; visualization, I.D.; supervision, N.S. and T.V.; project administration N.S.; funding acquisition, N.S. All authors have read and agreed to the published version of the manuscript.

**Funding:** This research acknowledges the Signature award funding and startup funds from University of Arkansas at Little Rock.

**Institutional Review Board Statement:** Not applicable.

**Informed Consent Statement:** Not applicable.

**Data Availability Statement:** The data presented in this study are available on request from the corresponding author.

**Acknowledgments:** Authors acknowledge Taylor Scifres, Shawn Bourdo and Humam Shahare for their help in this project.

**Conflicts of Interest:** The authors declare no conflict of interest.

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
