# Peer review of "Effect of KOH on the Energy Storage Performance of Molasses-Based Phosphorus and Nitrogen Co-Doped Carbon"

_2673-3293, doi:10.3390/electrochem2010003_

Round 1

Reviewer 1 Report

The article presents an experimental study on the evaluation of the Effect of KOH on the Performance of Molasses-based Phosphorus and Nitrogen Co-doped Carbon. I think it is a well-written article and I would recommend publishing the article after addressing the following comments.

  • Line 39: authors mentioned, “The equation demonstrated that coal reserves will most likely last until the year 2112 and will be the only fossil fuel remaining after 2042”. This is not accurate as gas hydrates reservoir which is estimated to have at least 2 times more energy than other fossil fuels combined will stay remain after this point. Please see the following articles for more details and add a sentence about this in your article: doi.org/10.1039/C8CS00989A , doi.org/10.1021/acs.est.7b05784
  • Line 59: the following sentences does not have any references “These are often synthesized by chemical activation with an activating agent such as KOH, zinc chloride (ZnCl2), or H2SO4.

Traditionally this is a two-step process in which biomass is first carbonized and subsequently

activated.”

  • Introduction: introduction seems to be a general literature review. I would suggest adding a paragraph to explain the details of recent studies which are closely related to this work.
  • Methods: More results on electrochemical characteristics of the synthesized material could be conducted. Please add some other results on material conductivity, resistance (EIS), and so on. If you don’t have these results or can’t conduct additional experiments please explain in your article why they are not necessary for your study.
  • Figure 1 is not clear. Please add a magnified version of unclear part of the graph to the same figure. For example pore diameter 20-200.
  • Results and discussion: It would be valuable if authors discuss the effects of operating conditions such as ambient temperature on electrochemical performance of the developed electrode. The following paper can be helpful: ‘’ A review on temperature-dependent electrochemical properties, aging, and performance of lithium-ion cells’’
  • Figure 5: please change the scale of the y axis so the graph will be more clear. For example 100-250.

Reviewer 2 Report

This manuscript studies the effect of KOH activation of carbon electrodes synthesized from molasses on the electrochemical energy storage performance. However, it is unclear what the advancement of this work is compared to the other published works on activated carbon.  Why is their material unique? This must be highlighted in the introduction section. In addition, there are some issues that need to be addressed before further consideration.

  • There are some crystalline XRD peaks in Figure S1. Those peaks do not belong to carbon. What are they?
  • What gas was used for BET analysis/pore size analysis?
  • There is no difference in pore size distribution between PNDC1 and PNDC2. Only difference is very large peak at ~0 nm. It is not clear what the authors want to claim with this analysis.
  • Usually, KOH activation leads to larger surface area. But, their results show the opposite. Can authors provide some comments on this?
  • The composition analysis results from EDS and XPS are very different. Why are they so different?
  • Explanation of possible reaction scenario of how KOH increases O content is not clear. Better explanation is required.
  • The authors must show the XPS results in the main part. In addition, XPS fitting is really bad, especially for C and N. The authors need to make more efforts for XPS analysis because they want to use XPS for quantification of composition.
  • In the introduction part, the authors must show the previous works on doping effects and surface area effect on activated carbon for capacitors. There are many and many papers published. And they need to show what the novelty of this work is.  

Round 2

Reviewer 2 Report

Most of the concerns are well addressed. However, two important questions are not properly addressed:

1. There are some crystalline XRD peaks in Figure S1. Those peaks do not belong to carbon. What are they? The authors just claim the sharp peaks belong to crystalline graphitized carbon without physical background. While they provide some references, those papers did not provide any evidences why the sharp peaks are from graphitized carbon although those peaks are not matched with graphite or ordered carbon materials. The sharp peaks could be impurities in the samples. The authors need to clarify their argument with evidences or admit they could be impurities.

2. Explanation of possible reaction scenario of how KOH increases O content is not clear. Better explanation is required. The authors cited references and tried to explain the reaction process, but it is really vague. What degradation of KOH means? To What? What complexes do they mean? How does K intercalation into the carbon make oxygen-rich functional groups? Even the references they used do not show clear explanation. 
